

# Large language models and their impact in medical imaging education

Jiajia Zhu[1,2] and Huanhuan Cai[1,2]

[1] Department of Radiology, The First Affiliated Hospital of Anhui Medical University, Hefei, China

[2] Research Center of Clinical Medical Imaging, Anhui Province, Hefei, China

## ABSTRACT

The recent emergence of large language models (LLMs), which have unprecedented capabilities to process, analyze, and synthesize complex medical information with remarkable proficiency, is poised to have a disruptive impact on health care. In the field of medical imaging, LLMs can be applied with promise in generating radiology reports along with detecting and correcting errors, explaining medical imaging findings, indicating differential diagnoses based on imaging patterns, and providing recommendations on imaging modality and protocol selection. In parallel, LLMs could offer innovative solutions for individualized learning, intelligent tutoring, content generation, and clinical decision support in medical education. However, challenges such as incorrect responses, negative influence on critical thinking, academic integrity concerns, bias, and privacy issues must be addressed to ensure safe and effective implementation of LLMs. This review summarizes the current applications, potential benefits, inherent limitations along with appropriate mitigation strategies, and future directions of LLMs in medical imaging education, emphasizing the need for responsible integration to maximize their utilities while mitigating risks.

## INTRODUCTION

In recent years, the emergence of large language models (LLMs) marks a promising new era in health care technology. A systematic review indicates that LLMs such as OpenAI's GPT-4 and Google's Bard models trained on rich amounts of data represent a transformative development in artificial intelligence (AI) with profound implications for medicine and medical education (*Lucas, Upperman & Robinson, 2024*). These sophisticated models, built on transformer architectures, have demonstrated unprecedented capabilities in natural language understanding and generation, allowing to process, analyze, and synthesize complex medical information with remarkable proficiency (*Zhou et al., 2023*). The ubiquity of these LLMs poses a myriad of potential clinical uses, *e.g.*, providing individualized learning tools and aiding in decision-making.

More recent reviews suggest that the field of medical imaging may be a primary beneficiary of LLMs (*Bradshaw et al., 2025*; *Liu et al., 2025a*). For example, LLMs can detect and correct common radiology report errors (*Gertz et al., 2024*), explain medical imaging findings at a reading level and in a suitable way for patients (*Berigan et al., 2024*),

Corresponding author
Huanhuan Cai, cjhl121@163.com

and indicate differential diagnoses based on medical history and imaging patterns of patients (*Kottlors et al., 2023*). It is also known that LLMs can outperform medical experts in clinical text summarization and create concise clinical summaries to guide radiologists (*Van Veen et al., 2024*), as well as assist in the determination of radiologic study and protocol based on request forms (*Gertz et al., 2023*). These highly promising applications suggest that LLMs could have a considerable impact on many aspects of clinical medical imaging.

The rapid advancement of LLM technologies also come at a time of rapid evolution in medical education (*Buja, 2019*). In the field of medical imaging education, LLMs hold immense promise to offer innovative solutions to longstanding challenges in radiology training, learning, and practice. The traditional model of medical imaging education, which combines didactic lectures, case-based learning, and hands-on interpretation under supervision, faces increasing pressure from growing imaging volumes, curriculum demands, and the need for continuous professional development. LLMs present opportunities to improve this educational paradigm through offering individualized learning tools and intelligent tutoring systems. It is evident that LLMs like GPT-4 can achieve passing scores on medical licensing examinations and perform comparably to human experts in answering radiology board-style questions. More specialized models fine-tuned on radiology reports have shown even greater promise for domain-specific applications. Nevertheless, integrating these LLMs into medical imaging education might raise concerns about accuracy, ethical implications, and detriments to critical thinking.

This comprehensive review examines the current knowledge on LLM applications in medical imaging education, exploring their potential benefits, inherent limitations, and future directions. We analyze how these models are being implemented across various educational contexts and discuss the technical, ethical, and practical considerations surrounding their adoption. The audience it is intended for radiology trainees, medical educators, medical students, practicing radiologists, and the relevant researchers.

## LITERATURE SEARCH METHODOLOGY

We performed literature searches across four major electronic databases: PubMed (MeSH), MedLine, Google Scholar, and Web of Science. The search strategy combined keywords and Boolean operators to capture relevant studies. Key search terms included: "large language models", "LLM", "ChatGPT", "GPT-4", "artificial intelligence", "medical imaging", "radiology", "education", "medical education", and their combinations. The search covered publications from January 2010 to June 2025, with a focus on recent advances from 2022 onward, coinciding with the widespread adoption of generative LLMs. We included original research articles, review articles, conference proceedings, and influential preprint studies (*e.g.*, from arXiv) that discussed LLM applications in medical imaging or related educational contexts. We excluded studies that: (1) did not focus on LLMs or their applications in medical imaging or education; (2) were not available in English; (3) were purely technical reports without clinical or educational relevance; and (4) focused exclusively on non-medical domains. We adopted a narrative synthesis framework to organize and summarize findings thematically. Given the rapidly evolving

nature of the field and the diversity of study types, a narrative approach was deemed most appropriate to integrate evidence from both empirical studies and conceptual reviews. Key themes were identified iteratively, and findings were critically evaluated to balance both optimistic and cautionary perspectives.

## DEVELOPMENT OF LLMS IN MEDICAL IMAGING

The development of LLMs in medical imaging consists of two stages: self-supervised pretraining and fine-tuning.

During the pretraining stage, massive amounts of unlabeled text data are used to teach the model the rules and patterns of language, achieved *via* self-supervision. For LLMs designed for use in medical imaging, datasets used for self-supervised pretraining should include radiology reports, clinical notes, and structured data from electronic health records (*Silva & Matos, 2022*). Some specialized LLMs, such as RadBERT (*Yan et al., 2022*) and Radiology-Llama2 (*Liu et al., 2023d*), could outperform general-purpose LLMs in clinical radiology tasks (*Zhang et al., 2024*). Due to the fact that pretraining LLMs requires enormous clinical datasets and large computational resources, most academic groups have been precluded from developing their own state-of-the-art LLMs. After pretraining, a general model, which can produce coherent and relevant text, is created. Then, fine-tuning is necessary for adapting this preliminary pretrained model to specific tasks and domains.

Fine-tuning typically starts with preparing labeled data related to the target task. For example, if the objective is to build an LLM to summarize radiology findings into impressions, datasets of paired findings and impressions should be utilized. Fine-tuning also includes preference alignment, which helps make sure that the LLM is more helpful, ethical, and factual. For instance, after initial training, ChatGPT underwent reinforcement learning with human feedback. This procedure involved humans evaluating the model's responses and offering their preferences, which were utilized to create a reward function to reinforce desirable behaviors and suppress undesirable behaviors. Recent advances in few-shot and zero-shot learning have enabled LLMs to perform novel tasks with minimal examples, while techniques like retrieval-augmented generation have allowed the models to incorporate up-to-date medical knowledge from external databases.

## APPLICATIONS OF LLMS IN RADIOLOGY

In medical imaging, LLMs can be applied with promise in radiology reporting, medical record navigation, and clinical decision-making.

LLMs can automate and improve many reporting-related tasks. There has been extensive research on LLMs-based clinical text (including radiology report) summarization (*Liu et al., 2021*; *Ma et al., 2023*; *Sun et al., 2023*). It has been shown that LLMs can match the performance of experts in impression generation, clinical note summarization, and doctor-patient dialog summarization. This may help radiologists to generate more accurate reports, improving both efficiency and report quality. LLMs can also be leveraged to suggest differential diagnoses on the basis of radiology observations (*Kottlors et al., 2023*), to detect speech recognition errors in radiology reports (*Schmidt et al., 2024*), and to transform free-test radiology reports into structured reporting (*Adams et al., 2023*). While

preliminary and investigative research has yielded promising results, real-world clinical assessment is still warranted. Furthermore, personalizing reporting styles to individual hospital or radiologist (*Tie et al., 2024*) and integrating LLMs into clinical workflows remain challenging (*Kim, 2024*), such that careful model monitoring and human oversight are greatly needed.

The additional application of LLMs is to improve decision-making for clinical imaging settings, that said, they may enhance clinicians' abilities and efficiency in decision-making. For instance, LLMs can provide recommendations on imaging modality and protocol selection according to the patient's medical history and the referring clinician's questions (*Gertz et al., 2023*; *Liu et al., 2022*). While numerous studies have demonstrated that LLMs are able to achieve performance levels comparable to human experts in recommending imaging strategies for distinct clinical manifestations, some concerns still exist regarding adapting to institution-specific guidelines and handling multiphasic explanations, implying that continued validation in more clinical datasets or customized fine-tuning is necessary.

Considerable effort in the past decade has been directed towards developing LLM tools specializing for medical imaging. For example, CheXbert (*Smit et al., 2020*) and RadGraph (*Jain et al., 2021*) could automate the extraction of important clinical information from unstructured radiology reports. In recent years, more updated LLM tools have shown promising results in extracting abnormal observations (*Guellec et al., 2024*), delineating lesion features (*Fink et al., 2023*), and assessing treatment efficacy (*Huemann et al., 2023*), which may facilitate longitudinal studies and retrospective investigations.

Beyond text-based applications, vision-language models (VLMs) represent a transformative advancement by directly interpreting medical images alongside textual queries. These multimodal systems enable interactive educational applications, such as visual question answering and anomaly localization, allowing trainees to engage in dynamic dialogues about findings within actual imaging studies (*Liu et al., 2023a*). While challenges like visual hallucinations require careful mitigation, VLMs hold significant potential for creating immersive, image-centric learning environments that closely mimic clinical reasoning workflows.

# APPLICATIONS IN MEDICAL IMAGING EDUCATION

The applications of LLMs in medical imaging education are multifactorial (Fig. 1).

## Performance on standardized examinations

For example, medical education could benefit from LLMs like ChatGPT, which have shown competence in multiple standardized clinical exams involving neurosurgery (*Ali et al., 2023*), ophthalmology (*Antaki et al., 2023*), plastic surgery (*Humar et al., 2023*), general surgery (*Oh, Choi & Lee, 2023*), clinical toxicology (*Sabry Abdel-Messih & Kamel Boulos, 2023*), cardiology (*Skalidis et al., 2023*), Medical Licensing Exam (*Kung et al., 2023*), and Medical Final Examination (*Rosol et al., 2023*), rendering them a potentially valuable reference for students and educators (*Liu et al., 2023b*). For medical students who need to learn a large body of medical knowledge during a short period, LLMs offer direct

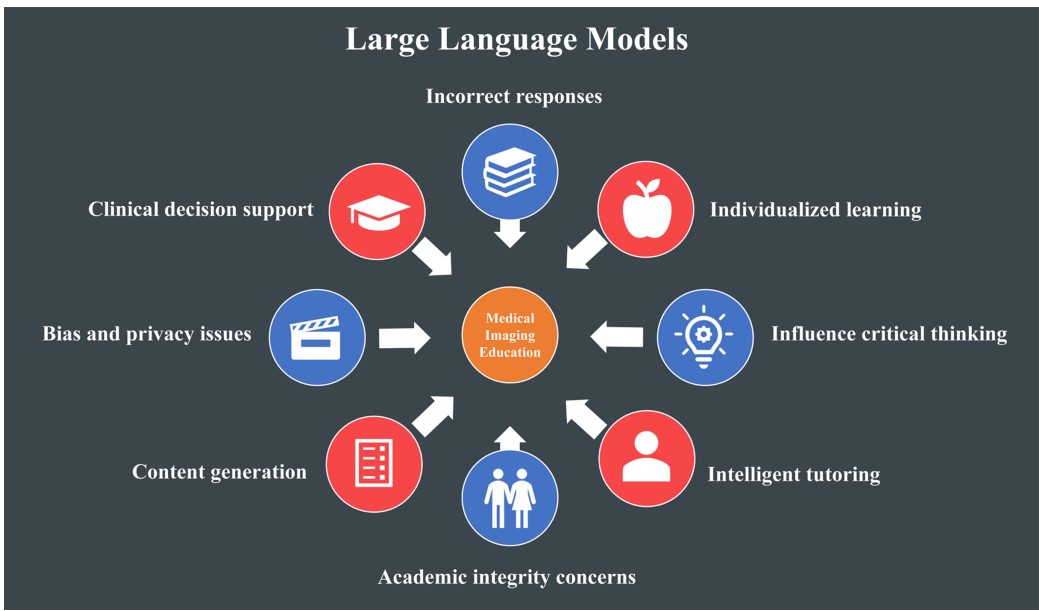

**Figure 1 Applications and limitations of large language models (LLMs) in medical imaging education.** The applications have a red icon and the limitations have a blue icon.

access to medical resources, research literature, and clinical guidelines. From a high-level point of view, with immediate access to answers on how to manage classical or even complicated clinical scenarios at the fingertips of students and clinicians, it raises the question concerning what factually is the fundamental medical knowledge and understanding required for medical students before advancing to a higher educational level. While it is still not clear how the utilization of LLMs will assist medical educators, LLMs may indeed help educators create tests with improved ability to evaluate medical students' knowledge in realistic settings. Nonetheless, exam performance level alone may not always indicate the clinical knowledge of a student. Likewise, satisfactory exam performance of LLMs does not always suggest the usefulness of these AI techniques in the education of future radiologists.

## AI as interactive teaching assistants

In medical education and training, LLMs have the potential as a powerful tool to advance conventional coursework and teaching, providing medical students with realistic clinical scenarios and feedback (*Webb, 2023*). For radiology training, LLMs can be used to detect discrepancy between preliminary trainee reports and final attending reports, which can facilitate the identification of difficult teaching cases. Furthermore, GPT-4 has been demonstrated to yield high-quality case reports and show proficiency in radiology. From a literacy viewpoint, LLMs can assist students and educators in ensuring writing accuracy, standard style and formatting, and appropriate language and terminology, suggested by recent reviews (*Abd-Alrazaq et al., 2023*; *Gandhi Periaysamy et al., 2023*). In addition, LLMs can help medical students in performing comprehensive evaluation of literature and

synthesizing the information in a readily comprehendible way (*Liu et al., 2023c*; *Parsa & Ebrahimzadeh, 2023*). Some LLM functions such as gathering and organizing information can help save time for busy medical students and offer learners real-time feedback on clinical experiences.

## Personalized tutoring and chatbots

With respect to educators, LLMs are beneficial for educating radiology trainees. For instance, LLM-powered chatbots represent useful instruments for interpreting complex medical concepts (*Singhal et al., 2023*), simplifying radiology reports (*Amin et al., 2023*), and resolving questions about radiologic procedures (*Rahsepar et al., 2023*). Nevertheless, there have been several concerns regarding the accuracy and completeness of the provided information, with further studies greatly needed in the future. In general, LLMs facilitate teaching by offering curriculum and evaluation planning as well as support educators and their administration by allowing to allocate resources in a more effective manner (*Shorey et al., 2024*). LLMs provide instant feedback and support along with interactive learning experiences and scalability for educators. Moreover, by enhancing education efficiency, LLMs might help address the shortage of medical educators, particularly in resource-limited circumstances, and offer more time for educators to focus their efforts on individual student mentorship.

## Enhancing clinical reasoning and problem-solving skills

LLM systems allow medical students to examine subjects in unprecedented detail and gain insights by exploiting massive amounts of datasets to address current problems. LLMs could improve clinical reasoning and elevate problem-solving abilities by permitting students to raise questions, receive explanations, and give hypotheses regarding disorders as well as their diagnoses and treatments (*Ahn, 2023*). Further, LLMs can be used for medical education early in non-clinical environments, combined with present resources like flashcards and practice question banks (*Singh et al., 2023*). Despite the promising applications of LLMs (*Sng et al., 2023*), they cannot replace the in-person medical education offered by traditional classes and clinical rotations.

Collectively, LLMs have provided large valuable resources for medical imaging students by offering immediate access to medical knowledge and research literature. While LLMs' performance has varied, further specialization could potentially support their functions, including preparing exams, generating case reports, supporting clinical care assessment, and improving clinical reasoning through answering medical questions and producing hypotheses. Nevertheless, some challenges and limitations remain concerning achieving human-like dialogue and interaction in their current forms. Table 1 summarizes the key applications, benefits, and limitations of LLMs across various domains of medical imaging education, providing a comprehensive overview of their current capabilities and constraints.

**Table 1 Applications, benefits, and limitations of large language models in medical imaging education.**

| Application domain | Specific examples | Key benefits | Main limitations |
|---|---|---|---|
| Individualized learning | • Customized learning paths<br>• Adaptive content delivery<br>• Knowledge gap identification | • Personalized pacing<br>• Tailored to learner's level<br>• Continuous availability | • Limited personalization depth<br>• May reduce peer interaction<br>• Requires extensive learner data |
| Intelligent tutoring | • Real-time Q&A support<br>• Step-by-step explanations<br>• Case-based guidance | • Immediate feedback<br>• 24/7 availability<br>• Consistent responses | • Potential for incorrect answers<br>• Lack of contextual understanding<br>• Limited emotional intelligence |
| Content generation | • Case report creation<br>• Examination questions<br>• Educational summaries | • Time-saving for educators<br>• Diverse content types<br>• Standardized formatting | • Quality variability<br>• Potential factual errors<br>• Requires expert validation |
| Assessment & evaluation | • Automated scoring<br>• Performance analytics<br>• Progress tracking | • Objective assessment<br>• Detailed feedback<br>• Efficient evaluation | • May not capture clinical nuance<br>• Limited practical skill assessment<br>• Academic integrity concerns |
| Academic writing support | • Literature synthesis<br>• Manuscript drafting<br>• Language polishing | • Improved writing efficiency<br>• Standardized terminology<br>• Grammar correction | • Plagiarism risks<br>• Loss of original thought<br>• Ethical disclosure requirements |

## CHALLENGES AND LIMITATIONS

Despite their potential, LLMs in medical imaging education face several significant challenges and limitations: incorrect responses, negative influence on critical thinking, academic integrity concerns, bias, and privacy issues (Fig. 1). Although these challenges could limit their usability, appropriate strategies can help mitigate these concerns.

LLMs could yield incorrect responses, referred to as hallucinations or fact fabrications. This is particularly common for applications in the field of medical imaging, where the context of specific medical language is crucial and decisions are high stakes. Thus, it is necessary to validate the accuracy of information provided by LLMs and continue to highlight practical skills. The optimization of LLMs for specific use cases could decrease incorrect responses. For example, LLMs can be connected to specific internet databases (*e.g.*, PubMed) to ground them in up-to-date information (*Min et al., 2024*). In radiology, domain-specific models such as RadBERT (*Yan et al., 2022*) and Radiology-GPT (*Liu et al., 2025b*) have shown greater promise.

Experts have posited concerns regarding the negative influence of overreliance on LLMs on clinical reasoning development and the lack of contextual thinking in LLMs' responses (*Abd-Alrazaq et al., 2023*). In addition, there have been issues regarding redundancy and a lack of original thoughts, which might adversely impact students' reasoning skills and critical thinking (*Arif, Munaf & Ul-Haque, 2023*; *Gunawardene & Schmuter, 2024*; *Huh, 2023*). It has been traditionally assumed that LLMs show poor complex reasoning. Although more recent LLMs have made progress, complex multistep reasoning is greatly required for better applications in radiology and its education. For instance, during the

model fine-tuning procedure, multistep reasoning can be achieved through giving feedback on each reasoning step in a chain of thoughts rather than giving feedback solely on final answers (*Lightman et al., 2023*).

With regard to academic integrity concerns, it is largely known that organizational guidelines have been modified due to LLMs' ability to produce articles (*Loh, 2023*). Other concerns involve fabricated references and false information perpetuation (*De Angelis et al., 2023*; *Gravel, D'Amours-Gravel & Osmanlliu, 2023*; *Perera Molligoda Arachchige, 2023*). Several experts even advocate halting improved LLM training in light of societal concerns. Some more tempered suggestions implicate thorough testing, supervised trainee exploration, and open discussions around responsible and appropriate use of LLMs in a recent review (*Bair & Norden, 2023*). Therefore, it is critical for medical imaging students to obtain a complete understanding of LLMs' appropriate and effective use as a powerful resource to address conventional challenges, as early as the period of preparing medical school application (*Hashimoto & Johnson, 2023*; *Munaf, Ul-Haque & Arif, 2023*). In both medical research and education, the utilization of LLMs in article or project preparation might raise ethical issues and therefore clarity concerning appropriate methods of disclose is warranted. Researchers should claim the extent to which LLMs are engaged in their project or article preparation, since transparency is the best approach to combating resistance to their use. Advancements in the knowledge about the LLM techniques would be improved with transparency concerning the utilization of LLMs.

Another major concern for the use of LLMs is the potential for biases in trained models (*Chowdhury, 2023*). In case of biased training datasets or algorithms themselves, a risk emerges that LLMs are prone to perpetuate biases present in their training data or adopted algorithms, which may result in inaccurate information delivered to medical students as well as a lack of diversity and representation in medical education (*Temsah et al., 2023*; *You et al., 2025*). These biases could ultimately have adverse impacts on learning and patient care in medical imaging education. The above-described approaches used for the optimization of model performance could also be leveraged to decrease biases. For example, reducing biases in training data and fine-tuning procedures with use of reinforcement learning from human feedback are likely to be able to lower the risks.

Privacy and confidentiality constitute key considerations in the implementation of LLMs into medical imaging education because sensitive information, such as patient clinical and research data, is often shared when using LLMs. It is generally accepted that patient data are protected by the Health Insurance Portability and Accountability Act (HIPAA). The applications of LLMs in medical imaging education settings must ensure that patient information should not be breached and privacy is maintained. Some privacy-preserving LLMs, including those that are used locally, can protect personal health information from third parties (*Cai, 2023*; *Mukherjee et al., 2023*). Researchers, educators, and students can access some LLMs securely through HIPAA-compliant application programming interfaces, whereby users possess the input and output datasets. Entry into sensitive patient data should be avoided and could only be considered after receiving review board approval and consulting institutional guidelines.

While LLMs are largely beneficial in medical imaging education, responsible implementation *via* continued validation, emphasizing practical skills, monitoring for ensuring accuracy, and mitigating potential risks is still critical and necessary.

## CURRENT STATE AND FUTURE OUTLOOK

It is already evident that rapid progress in LLMs has sparked excitement linked to applications in medical imaging and its education, with the adoption of LLMs in medical imaging education evoking a rich range of reactions among educators, learners, and clinicians. This review fills in a missing gap in current literature on LLMs in medical imaging education by extending beyond the primary focus on the possible misuse of LLMs in education to the widespread investigation of their transformative potential. With an increase in size and sophistication, state-of-the-art LLMs are expected to improve in performance. Creating and assessing models optimized for specific applications would aid in shaping how LLMs are employed to improve efficiency, cost-effectiveness, and outcomes in medical imaging education. However, it is notable that LLM systems are by no means a replacement for traditional education and experience, since they still fall short in several aspects of medical knowledge and application in comparison to human performance.

Educators, students, or clinicians who are prone to welcome LLMs typically recognize their potential utility to mitigate concerns in the field of medical education, such as the overload of medical curricula as well as the everchanging knowledge with growing research. The rich range of information might prevent students or clinicians from being experts in all aspects of medicine. It is largely accepted that LLMs could provide an approach to enhancing the learning processes, especially focusing on the most critical aspects needed at each certain level of medical training. Additionally, LLM systems are able to render medical knowledge more accessible since they offer more widespread access to the latest medical research information, thereby decreasing barriers and reconciling disparities in medical education. This is particularly beneficial for individuals who have relatively limited access to costly printed or online up-to-date medical resources, since they could derive updated medical knowledge readily and at minimal cost with use of LLMs.

LLMs could offer tailored explanations, practice questions and feedback that is able to provide more individualized learning approaches suitable for each student. If students obtain and grasp complex medical knowledge more quickly and efficiently in a personalized manner, the overall efficiency of the medical education process would be enhanced. Moreover, LLM systems could impose a beneficial influence on the development of clinical reasoning, critical thinking and decision-making skills, by offering a safe and interactive circumstance for students to engage them in formulating, analyzing and discussing real patient cases.

Despite the above-described applications, refusal to adopt LLMs arises from issues regarding their tendency to provide incorrect information, the ethical concerns about their utilization and reliability. In addition, there is also fear towards the utilization of LLMs resulting in a less emphasis on the development of critical thinking and decision-making skills among medical learners. Furthermore, there is apprehension regarding data privacy

and the lack of personalized education as well. It is possible that the extent of familiarity to the LLM systems would influence comfort and understanding profiles. Individuals who have more exposure to LLM approaches are more prone to understand the tools' present abilities and challenges. For LLMs, the tendency to hallucinate people or generate actually false information gives another element of anxiety among educators and students. Hallucinations are particularly problematic in the field of medical education, as accuracy is paramount for medical education. Thus, it is necessary to conduct critical evaluation and validation of LLMs' outputs. Integration of these LLM techniques in a manner that improves medical education necessitates mitigation of these concerns in a fashion that is forward-thinking and collaborative across multiple disciplines to ultimately transform educational strategies and ethical guidelines.

While educators and students have increasingly recognized LLMs' potential to contribute to the revolutionization of medical education, there is still apprehension regarding their reliability and adverse influence on learners' critical thinking skills. To address these challenges, it seems necessary to develop mechanisms for ensuring information accuracy and implementing LLMs in a manner that complements and extends, rather than replaces, conventional teaching approaches. In order to harness the profits of LLMs in medical imaging education, it is important to use them responsibly, which involves rigorous monitoring to validate the accuracy of information offered by LLMs. As accuracy is paramount in clinical care, the future generations of clinicians must be provided with more reliable tools to improve their learning. The implementation of LLMs into medical imaging education would offer opportunities to improve curricula rather than detract from it. That said, scientific and proper applications of LLMs would make students to place less efforts on memorization and building of mnemonics for test-taking, and instead to focus more on comprehension of knowledge, logical thinking, problem-solving and the application of medical concepts into clinical practice.

Critically, the utilization of LLMs must continue to ensure patient data privacy and confidentiality in health care settings. These priorities should be repeatedly highlighted when learners use these LLM techniques. Collaboration between technical experts and clinicians to align priorities and ensure that the development of these updated tools meets the actual needs of medical education is quite necessary. For the medical education community, learning experiences can be improved while concerns of the LLM technologies can be mitigated if users understand and master the factors that influence the applications of LLMs.

## CONCLUSION

Large language models (LLMs) have rapidly evolved and demonstrate significant potential to enhance performance, improve efficiency, and create value in medical imaging education. They offer innovative solutions for individualized learning, intelligent tutoring, content generation, and clinical decision support. While these applications are promising, key challenges—including ethical concerns, information inaccuracy, privacy issues, and potential impacts on critical thinking—must be responsibly addressed before widespread adoption.

Future research should focus on the following directions:

- **Multimodal integration:** Developing and evaluating vision-language models that combine image interpretation with textual analysis to better support radiology education.
- **Curriculum integration frameworks:** Designing structured approaches for embedding LLMs into existing medical imaging curricula while maintaining educational integrity.
- **Personalized learning pathways:** Creating adaptive LLM systems that respond to individual learner progress, knowledge gaps, and preferred learning styles.
- **Critical thinking and clinical reasoning:** Investigating how LLM use influences the development of diagnostic reasoning and problem-solving skills in trainees.
- **Ethical and bias mitigation strategies:** Establishing guidelines and technical solutions to reduce model bias, ensure fairness, and protect patient privacy.
- **Long-term outcome studies:** Conducting longitudinal and controlled trials to assess the real-world educational impact of LLMs on radiologist training and patient care outcomes.

As LLM technology continues to advance, collaborative efforts among educators, clinicians, and AI developers will be essential to harness its benefits while mitigating risks. With responsible implementation and ongoing research, LLMs are poised to become valuable assets in the future of medical imaging education.

### Funding
This work was supported by the National Natural Science Foundation of China (82471952), Anhui Provincial Natural Science Foundation (2308085MH277), Scientific Research Key Project of Anhui Province Universities (2022AH051135) and the Scientific Research Foundation of Anhui Medical University (2022xkj143). The funders had no role in study design, data collection and analysis, decision to publish, or preparation of the manuscript.

### Grant Disclosures
The following grant information was disclosed by the authors:
National Natural Science Foundation of China: 82471952.
Anhui Provincial Natural Science Foundation: 2308085MH277.
Scientific Research Key Project of Anhui Province Universities: 2022AH051135.
Scientific Research Foundation of Anhui Medical University: 2022xkj143.

### Competing Interests
The authors declare that they have no competing interests.

## Author Contributions

- Jiajia Zhu performed the experiments, analyzed the data, performed the computation work, authored or reviewed drafts of the article, and approved the final draft.
- Huanhuan Cai conceived and designed the experiments, prepared figures and/or tables, authored or reviewed drafts of the article, and approved the final draft.

## Data Availability

This is a literature review.

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
