# Peer review of "Large language models and their impact in medical imaging education"

_PeerJ Computer Science, doi:10.7717/peerj-cs.3433_

## Round 0.1 · original submission · Major Revisions

· Academic Editor

Major Revisions

**Language Note:** When preparing your next revision, please ensure that your manuscript is reviewed either by a colleague who is proficient in English and familiar with the subject matter, or by a professional editing service. PeerJ offers language editing services; if you are interested, you may contact us at [email protected] for pricing details. Kindly include your manuscript number and title in your inquiry. – PeerJ Staff

Reviewer 1 ·

Basic reporting

Language and structure:
The manuscript is generally well-structured. The language is professional; however, it contains repetitive phrases (e.g., "LLMs hold immense potential") and occasionally dense sentence constructions. Simplifying these sections and reducing redundancy would improve overall readability. A final review by a native English speaker or professional editing service is recommended.

Literature and citations:
The manuscript references numerous recent studies (2023–2025), indicating a strong command of current literature. However, citation clusters (e.g., lines 136–139) are overly dense, making it unclear which study supports which claim. Additionally, several citations refer to opinion pieces or narrative reviews; these should be clearly distinguished from empirical studies within the text.

Figures and tables:
While the single figure included is relevant, the paper would benefit from additional visual aids—such as summary tables comparing LLM capabilities or outlining key advantages/disadvantages in medical imaging education.

Scope and novelty:
The manuscript addresses a multidisciplinary and emerging topic at the intersection of computer science, medical education, and radiology. Although LLMs in medical education have been widely discussed recently, this paper offers a novel and focused perspective on medical imaging education, which justifies its contribution.

Experimental design

The manuscript is a narrative review, and this is an appropriate format for the topic. However, the methodology section is too limited (lines 70–75). Key methodological elements are missing, including:

Timeframe of the literature search

Inclusion/exclusion criteria

Number and type of studies included

Critical evaluation or synthesis approach

Structured or semi-structured review framework (e.g., PRISMA, scoping review model)

Without this information, it is difficult to assess the comprehensiveness or objectivity of the review. While the article does not claim to be systematic, the authors should adopt a more transparent and structured approach in revising this section.

Validity of the findings

The main arguments are generally well-supported by the cited literature. The authors balance discussions of LLMs’ strengths with acknowledgment of limitations. However:

Some claims (e.g., that LLMs can enhance clinical judgment) are speculative and should be softened using tentative language (e.g., "may improve," "potentially support")

Future research needs (e.g., integration into curricula, personalization, critical thinking outcomes) are briefly mentioned but should be more clearly summarized—preferably in bullet form in the conclusion.

The inclusion of more empirical studies, particularly those assessing real-world applications of LLMs in radiology education, would enhance the credibility of the findings.

Additional comments

Strengths:

Timely and relevant topic with multidisciplinary value

Covers technical, pedagogical, and ethical dimensions

Strong use of recent literature

Weaknesses and suggestions:

Language repetition and stylistic density detract from clarity.

Methodological transparency needs improvement.

Structural reorganization is needed, particularly within the "Medical Imaging Education" section, which is overly long and unfocused.

Suggested restructuring with subheadings:
-Assessment and Exam Performance in Radiology Education
-AI-Based Question Generation and Psychometric Evaluation
-AI as a Learning Assistant and Feedback Provider

Conclusion section:
Currently lacks a strong summary of the paper’s contribution and future directions. A short, clear list of remaining research gaps would help solidify the paper’s value.

Cite this review as

Reviewer 2 ·

Basic reporting

This paper provides a comprehensive literature review on the applications, opportunities, challenges, and future prospects of large language models (LLMs) in medical imaging education. The authors explain how LLMs, especially models such as GPT-4, can support radiology trainees, medical educators, and clinicians by generating educational content, assisting in clinical decision-making, and enhancing personalized learning. In addition to highlighting key benefits, the article addresses important concerns, including hallucinations, bias, academic integrity, and privacy issues, and proposes strategies to mitigate them.

Strengths of the paper:
+ The paper presents a broad overview of LLM applications across both radiology practice and education, supported by a wide range of recent publications.

+ The authors incorporate numerous up-to-date references from 2023 to 2025, which reflect the fast pace of advancement in this field.

+ The manuscript carefully considers both the promises and limitations of LLMs, offering a responsible and realistic view of their medical imaging educational value.

Weaknesses of the paper:
- Since radiology relies heavily on visual data, the absence of discussion about vision-language or image-based LLMs limits the completeness of the review.

- The paper would benefit from the inclusion of additional figures or visual examples. Aside from the single diagram summarizing applications and limitations, there are no accompanying images, case illustrations, or conceptual diagrams to support the text. Adding such visuals could improve reader engagement and clarify key concepts, especially for a topic related to medical imaging.

- The authors seem to have missed some relevant literature. Specifically, they don't discuss learning-based methods for medical imaging tasks, missing out on several relevant citations, e.g. "Application of large language models in medicine", "Uncovering Memorization Effect in the Presence of Spurious Correlations", "Benchmarking Large Language Models on CMExam - A comprehensive Chinese Medical Exam Dataset", "A medical multimodal large language model for future pandemics", "A survey of large language models in medicine: Progress, application, and challenge", "Auto-encoding knowledge graph for unsupervised medical report generation", "Retrieve, reason, and refine: Generating accurate and faithful patient instructions", "UniHGKR: Unified Instruction-aware Heterogeneous Knowledge Retrievers". These methods are relevant to the method proposed in this paper. These relevant papers should be included in the reference list.

Experimental design

-

Validity of the findings

-

Cite this review as

---

## Round 0.2 · accepted · Accept

· Academic Editor

Accept

The authors' have addressed all of the reviewers' comments. This manuscript is ready for publication.

Reviewer 1 ·

Basic reporting

The revised version of the manuscript has been carefully reviewed. The authors have adequately addressed the comments and suggestions raised in the previous round of review. The revisions have improved the clarity, coherence, and overall quality of the paper, particularly in the methodology and results sections.

In its current form, the manuscript meets the publication standards of the journal. Apart from minor editorial or stylistic corrections, no further revisions are deemed necessary.

Experimental design

no comment

Validity of the findings

no comment

Additional comments

no comment

Cite this review as

Reviewer 2 ·

Basic reporting

Thank the authors for the classification. It addressed all my concerns.

Experimental design

no comment

Validity of the findings

no comment

Additional comments

no comment

Cite this review as